# Pharmacological Insights into Halophyte Bioactive Extract Action on Anti-Inflammatory, Pain Relief and Antibiotics-Type Mechanisms

**DOI:** 10.3390/molecules26113140

**Published:** 2021-05-24

**Authors:** Rocco Giordano, Zeinab Saii, Malthe Fredsgaard, Laura Sini Sofia Hulkko, Thomas Bouet Guldbæk Poulsen, Mikkel Eggert Thomsen, Nanna Henneberg, Silvana Maria Zucolotto, Lars Arendt-Nielsen, Jutta Papenbrock, Mette Hedegaard Thomsen, Allan Stensballe

**Affiliations:** 1Department of Health Science and Technology, Aalborg University, 9220 Aalborg, Denmark; rg@hst.aau.dk (R.G.); zsaii14@student.aau.dk (Z.S.); tbgp@hst.aau.dk (T.B.G.P.); meth@hst.aau.dk (M.E.T.); nhenne16@student.aau.dk (N.H.); LAN@hst.aau.dk (L.A.-N.); 2Department of Energy Technology, Aalborg University, 9220 Aalborg, Denmark; mfred@et.aau.dk (M.F.); lssh@et.aau.dk (L.S.S.H.); mht@hst.aau.dk (M.H.T.); 3Center of Health Sciences, Department of Pharmaceutical Science, Federal University of Santa Catarina, Campus Universitário, Trindade, 88040–970 Florianópolis, Brazil; szucolotto@hotmail.com; 4Institute of Botany, Leibniz University Hannover, D-30419 Hannover, Germany; Jutta.Papenbrock@botanik.uni-hannover.de

**Keywords:** secondary metabolites, halophytes, hydroxycinnamic acid, inflammation, nutraceuticals

## Abstract

The pharmacological activities in bioactive plant extracts play an increasing role in sustainable resources for valorization and biomedical applications. Bioactive phytochemicals, including natural compounds, secondary metabolites and their derivatives, have attracted significant attention for use in both medicinal products and cosmetic products. Our review highlights the pharmacological mode-of-action and current biomedical applications of key bioactive compounds applied as anti-inflammatory, bactericidal with antibiotics effects, and pain relief purposes in controlled clinical studies or preclinical studies. In this systematic review, the availability of bioactive compounds from several salt-tolerant plant species, mainly focusing on the three promising species *Aster tripolium*, *Crithmum maritimum* and *Salicornia europaea*, are summarized and discussed. All three of them have been widely used in natural folk medicines and are now in the focus for future nutraceutical and pharmacological applications.

## 1. Halophyte Species for Current and Future Biomedical Applications

Salt-tolerant plant species, also called halophytes, belong to many botanical families. Halophytes are defined as plants that can grow and complete their life-cycle in a salt concentration of at least 200 mM NaCl [1]. They do not form a systematical group and, phylogenetically, they are not related to each other. Through evolution, they have adapted to saline conditions in several ways: morphologically, physiologically, and biochemically. Hence, halophyte species are found to produce high levels of bioactive compounds and free radical-scavenging secondary metabolites, potentially due to their adaptation to harsh environmental conditions. High salinity during growth and development also increases the level of free radicals in plants [2,3]. Such potential beneficial dietary factors in small doses and complex combinations (e.g., polyphenols, fibers, polyunsaturated fatty acids, etc.) for lifestyle changes can lead to reduced inflammation and improved health; however, metabolic disturbances are key contributors to disease progression [4]. Screening and testing of extracts from medicinal plants species, including halophytes, against a variety of pharmacological targets and disease conditions in order to benefit from the immense natural chemical diversity is a research focus in many laboratories and companies worldwide [5,6]. For halophytes, the relevant compound classes are a combination of components typical of lignocellulosic biomass and components unique for a family or species (Figure 1). The majority of the free radical-scavenging phenolic compounds are biosynthesized through the shikimic acid, acetic acid, and phenylpropanoid pathways, resulting in phenylpropanoids, simple phenols, and phenolics, respectively [7,8].

Over time, nature has been a rich source for various natural bioactive substances that have many applications in folk medicine and ethnopharmaceuticals [10,11,12]. For thousands of years, plants were the only source of medicine due to the absence of other compounds available, and they were used for a broad spectrum of medical purposes as therapies based on their contents of secondary metabolites and their bioactive properties [13].

The present review will focus on reviewing the current state of the art for biomedical applications of bioactive molecules present in three halophytic plant species, *Aster tripolium* (Jacq.) Dobrocz., *Crithmum maritimum* L., and *Salicornia europaea* agg. (illustrated in Figure 1), due to their potential use in nutraceutical foods, cosmetics, and also as bioactive components for medicinal applications because they contain health-promoting compounds such as minerals, fibers, oils, phenolics and vitamins [14]. The biological qualities of these three halophytic plants species can overall be divided into anti-inflammatory, antioxidant-rich and antibacterial activities, and future use in medicinal practice. The bioactive compounds are listed in Table 1 and summarized in Section 2. The chemical structure and biological activities of secondary metabolites isolated from *Salicornia europaea* L. have recently been reviewed [15].

## 2. Bioactive Components Including Primary Metabolites, Phenolics and Their Antioxidant Properties

The most common bioactive components include primary metabolites such as amino acids, proteins, bioactive polysaccharides, lipids, and lignin [29]. Dietary fibers and minerals (Mg, Ca, Fe, K) are also present in large amounts [3]. Immunomodulatory proteins, peptides, and polysaccharides have been isolated and characterized from *Salicornia* spp., that may explain some of the therapeutic efficacies which have been used in folk medicine to treat various diseases, including cancer [30]. The class of secondary metabolites or phytochemicals include compounds of pharmacological and biological importance, including alkaloids, fatty acids, and lipids, flavonoids, phenolics, quinines, tannins, terpenoids, steroids and saponins, and coumarins. The content of the secondary metabolites may vary depending on the particular habitat where the plant grows. The secondary metabolites from plants are challenging to categorize and replicate in the industry because their metabolic pathways of synthesis, features, and mechanisms of action often have similarities and overlap as well as there being an incomplete understanding of synthesis. However, a possible classification is based on their biosynthetic pathways which include large molecule categories: (a) phenolics including phenolic acids, their derivatives, and flavonoids; (b) terpenes and steroids; and (c) alkaloids [10].

Antioxidants can be found in high quantities in different foods, such as vegetables, berries and fruits, because most plants contain phenolic compounds, which are secondary metabolites with health beneficial properties, proven in in vivo and in vitro studies [31,32,33,34,35,36]. The common structural characteristics all phenolics share are aromatic rings, hydroxyl groups and, commonly, electron double bonds. Phenolics have been extracted using ethanol, methanol, water, ethyl acetate, and dichloromethane by the use of simple solvent extraction under reflux, Soxhlet extraction, assisted ultrasound extraction, and assisted microwave extraction [37,38,39,40,41]. Different extraction methods and solvents have been shown to target the extraction of different phenolics [41,42]. Especially, phenolic monomers have been shown to be thermolabile, and thereby prone to thermal degradation. Therefore, prolonged extractions using high temperatures should be avoided, and the extraction method should be chosen accordingly [38,43]. Rice-Evans et al. [43] investigated the impact of a number of hydroxyl groups on antioxidant activity of each molecule compared to vitamin E and proposed a correlation between the number of hydroxyl groups in HCAs, phenolic acids, flavonoids, their respective derivatives, and their antioxidant capacity. Here, the antioxidative capacity of the catechins epicatechin gallate (ECG) and epigallocatechin gallate (EGCG) is due to the molecular presence of seven and eight hydroxyl groups, respectively [44]. The latter compounds are well known antioxidants, relevant to cancer and neurodegenerative diseases [45,46].

Not only do a high amount of hydroxyl groups have high antioxidant activity, but Cano et al. [47] found the dimer of the hydroxycinnamic acid ferulic acid, 5–5′diferulic acid, to have a high superoxide anion free radical scavenging capacity compared to its monomer, ferulic acid. With a 35.7% inhibition of superoxide anion formation by 5–5′diferulic acid, compared to ferulic acid showing no direct inhibition of superoxide anion formation, Cano et al. [46] concluded that the type of linkage between ferulic acid monomers alters the superoxide anion free radical scavenging capacity, independent of the number of hydroxyl groups. The same trend of inhibition of superoxide anion formation after dimerization was also shown to be true for dimers of the flavonols kaempferol and quercetin containing multiple hydroxyl groups [47].

The beneficial properties of antioxidants are due to their ability to scavenge free radicals. Free radicals are atoms, molecules, or ions with unpaired electrons, and in biological systems are often derived from molecules containing oxygen, nitrogen, or sulfur, called reactive oxygen/nitrogen/sulfur species, or ROS, RNS, and RSS, respectively [48]. Common examples of ROS are the superoxide anion (O_2_^−^), hydroxyl (OH), and hydroperoxyl (HO_2_). Varieties of ROS are produced in living cells as products of oxygen utilization, with a possibility for these to lead to cellular damage, metabolic disorders, or cause DNA damage. Usually, enzymatic pathways can neutralize these free radicals, e.g., superoxide dismutase, catalase, or glutathione peroxidase [49]. However, these pathways might not be sufficient in all cases, and a non-enzymatic nutritional supplement of antioxidants is necessary.

Diet management using plant-based formulations may improve the metabolic status of patients, including those with diabetes mellitus, where an increased oxidative stress and chronic low-grade inflammation is observed as a consequence of the complex syndrome including long-term alterations of protein and lipid metabolisms [50]. Here, researchers found that a plant-based antidiabetic formulation including antioxidants was able to enhance total serum antioxidant defense and improve overall serum redox status and HDL redox function. More than 5000 flavonoids have been identified and are distributed in a wide range of plants; flavonoids possess documented anticancer activity, both in animal and cellular model systems [51]. Here, luteolin is an important natural antioxidant which has potent anticancer effects under both in vitro and in vivo conditions.

Flavonoids are a large family of polyphenolic compounds that consist of 12 major subclasses according to their chemical composition. Some of these are flavan-3-ols, flavonols, flavanones, flavones, isoflavones, and glycosylated flavonols [52]. These compounds provide an essential source of antioxidants in the human diet. They essentially exist in all foods, which have an origin from plants [20]. Although flavonoids and phenolics, in general, are thought to be non-nutritive agents, they are believed to have a possible health-promoting impact on multiple diseases such as cancer and atherosclerosis [20,53]. The nutraceutical value of *S. herbacea*-derived glucopyranosides as potent anti-obesity agents have been attributed to the alleviation of lipid accumulation [54]. Despite the broad range of key bioactive components, more clinical research is needed to substantiate the composition and quantity thereof in specific halophytes, as well as to determine the biomedical effects in health and disease.

## 3. Nutraceutical and Pharmacological Mode-of-Action of Key Secondary Metabolites in Halophytes

Wild edible plants (WEPs) are considered as promising sources of essential compounds, needed not only in the human diet including carbohydrates, proteins, and lipids, but also of other minor compounds such as phenols, vitamins, or carotenoids [55]. The presence of phenolic compounds in these vegetal matrices is supposed to provide a prophylactic effect against further pathogenesis and disorders related to aging or oxidative stresses. The utilization and valorization of phytochemicals have focused on nutraceutical use. Hence, the modes-of-action, pharmacological attributes, and medicinal properties target multiple common therapeutic areas, such as neuroprotective, antioxidant, analgesic, immunomodulatory, antimicrobial, antidiabetic and cardioprotective activities (Figure 2) [56]. Neuroprotection is attributed to high levels of antioxidants, including tungmadic acid, quercetin, and chlorogenic acid, enabling the scavenging of reactive oxygen species (ROS), e.g., H_2_O_2_, efficiently. These electrophilic compounds exert antioxidant activity as well as induce antioxidant enzymes through the Nrf2 signaling pathway, thereby exerting protective effects against ROS-induced neuronal cell damage [57]. Nrf2 regulation contributes to anti-inflammatory processes by orchestrating the recruitment of inflammatory cells, thus regulating gene expression through the antioxidant response element (ARE) [58]. NRF2 activation provides cytoprotection against numerous pathologies including chronic diseases of the lung and liver, autoimmune, neurodegenerative and metabolic disorders, and cancer initiation [59]. However, unidentified compounds may be co-responsible for the neuroprotective effect [60]. Linked to neuroprotection and ROS scavenging then HCAs and their derivatives also display antioxidant, anti-collagenase, anti-inflammatory, antimicrobial and anti-tyrosinase activities, as well as ultraviolet (UV) protective effects. This suggests that HCAs can be exploited as anti-aging and anti-inflammatory agents, preservatives and hyperpigmentation-correcting ingredients [61]. Recent findings suggest that the reversal of UVB-induced damages to skin may be prevented by the protecting effects of aqueous extracts of *S. europaea* affecting basal keratinocytes [62].

The involvement of *A. tripolium,*
*C. maritimum,* and *S. europaea* bioactive extracts in pain and itch mechanisms remains scarce, although several studies have been conducted, in vitro and in vivo, in order to demonstrate the effects of secondary metabolites present in the three species in relation to nociception and analgesia [63,64,65,66,67,68]. In a previous study, the authors looked at the analgesic action of chlorogenic acid (5-caffeoylquinic acid, CGA) in animal neuropathic pain models. CGA is a polyphenol formed by the esterification of caffeic and quinic acid, which can be found in plants, vegetables, and fruits [69], showing antioxidant, anti-inflammatory, antigenotoxic, anticancer, and cytostatic activities [63,70]. Dos Santos and colleagues [57] investigated the effect of pure CGA in the formalin-induced pain test, where CGA was reported to possess antinociceptive activity in neuropathic pain models, drastically reducing the pain behavior of mice after the injection of formalin [63]. Neuropathic pain could arise from tissue damage, inflammation, or injury to the nervous system, and is characterized by three sensory abnormalities which include increased sensitivity to painful stimuli (hyperalgesia), perception of innocuous stimuli as painful (allodynia), and spontaneous pain [64]. In this direction, another study showed the antinociceptive action of CGA in the neuropathic pain rat model [65]. The authors showed that the administration of CGA produces significant dose- and time-dependent anti-hyperalgesic effects in chronic constrictive nerve injury (CCI) rat models, and that a chronic treatment for 14 days reduces mechanical hyperalgesia in rats, suggesting action of the CGA on the inhibition of reactive oxygen species (ROS) [65]. Moreover, another study demonstrated that the administration of CGA systemically or intrathecally improves mechanical and cold hyperalgesia in rat neuropathic pain models [68]. In 2014, Qu et al. [61] suggested that the usage of CGA may exert analgesic action by modulating acid-sensing ion channels (ASICS) in rat dorsal root ganglion neurons [67]. The effect of plant extracts and secondary metabolites on itch or skin diseases is a topic that needs to be further evaluated, although a recent study looking into the effect of the prolonged application of *S. europaea*-based cream on sunburned skin of women showed that eight weeks of treatment can improve skin structure and texture, helping the recovery from topical induced sunburn [62].

## 4. Anti-Inflammatory and Antimicrobial Activities of the Three Halophytes

### 4.1. Characteristics and Medicinal Properties of Aster Tripolium

#### 4.1.1. Anti-Inflammatory Compounds of *Aster Tripolium*

*Aster tripolium* (syn. *Tripolium pannonicum*) is a halophyte that belongs to the Asteraceae family, which is one of the most consumed wild-gathered food in several European countries [71]. It is often found in coastal areas and sometimes in salty bogs [72]. It consists of multiple bioactive compounds, which gives *A. tripolium* considerable potential as a functional food ingredient. Apart from *A. tripolium* being highly useful for fodder and food, such as a salad or vegetable, it has high levels of nutrients in its leaves [73]. Additionally, these functional food ingredients may have a positive outcome on numerous diseases such as diabetes [74]. Some of the bioactive compounds are three types of caffeoyl esters which are isomers of CGA. Besides acting as anti-inflammatory and antihypertensive agents, as stated previously, evidence suggests that CGA from coffee possesses insulin sensitivity effects with the same mechanism of action as metformin [75]. Metformin is an antibiotic but has been approved as a first-line treatment for type 2 diabetes [76,77]. Glucose tolerance has been investigated in obese men, which revealed that treatment with CGA improved insulin responses. Both in vitro and in humans, researchers have demonstrated that CGA increases cell insulin secretion and thus glucose uptake [78,79]. These qualities make CGA valuable in the treatment of diabetes and obesity. Another bioactive compound found in *A. tripolium* at unreported levels is quercetin, which belongs to the flavonol subclass of flavonoids [53]. Quercetin is a very strong antioxidant, also found in foods such as apples, onions, and tea [16,20]. Accordingly, quercetin can chelate metals, scavenge oxygen-free radicals, and prevent the oxidation of low-density lipoprotein (LDL) in vitro [17]. Therefore, quercetin could thus contribute significantly to the antioxidant defenses present in blood plasma; it is reportedly able to inhibit the oxidation of LDL in atherosclerotic lesions and thereby be a natural anti-atherosclerotic diet component or be used in T2DM to achieve adequate glycemic control [80,81,82].

#### 4.1.2. Antimicrobial Compounds of *Aster Tripolium*

To the best of our knowledge, there is currently no evidence that suggests or clarifies whether *A. tripolium* possesses antimicrobial properties. Antibiotic resistance mechanisms are an increasing global health concern; therefore, research in this field with *A. tripolium* in mind is crucial. In general, antibiotics possess the ability to exert selective toxic or growth-limiting effects on bacteria. This selective toxic effect of the antibiotics is nontoxic for human cells but can simultaneously inhibit the functions and target the structures in the bacteria cell [83]. Antibiotics are commonly classified into bactericidal and bacteriostatic agents based on their antimicrobial action. The above-mentioned classification discriminates antibiotics that kill bacteria, referred to as bactericidal, and antibiotics that inhibit bacterial growth or reproduction, called bacteriostatic [84]. One way that bactericidal antibiotics kill bacteria is by inhibiting cell wall synthesis. Another mechanism of action includes the inhibition of key bacterial enzymes or protein translation. On the other hand, bacteriostatic antibiotics limit the growth of bacteria by interfering with bacterial protein production, DNA replication, or other aspects of bacterial cellular metabolism [85]. *A. tripolium* also contains CGAs, similarly to other halophytic species. CGA is a family of esters constructed between certain trans-cinnamic acids and trans-quinic acid [22]. Interestingly, however, if CGA isomers are hydrolyzed to quinic and caffeic acids, the latter have shown antimicrobial effectiveness against certain Gram-bacteria. Elegir et al. [18] demonstrated that caffeic acid revealed antibacterial effects against *Staphylococcus aureus*, and when increasing the concentration, the acid was also capable of exerting bactericidal activity against *Escherichia coli* [18,19]. According to multiple studies, caffeic acid has the strongest antibacterial effects observed when compared to other phenolic acids such as *p*-coumaric acid. It has been postulated that this might be due to one or more hydroxyl groups substituted at the caffeic acid phenol ring. Additionally, because caffeic acid is less polar, it is capable of exerting lipophilicity and thus impacts the permeability of the cell membrane of the bacteria and interferes with the aerobic metabolism. The cell membrane is crucial for the integrity of the bacterium, which explains why caffeic acid acts bactericidal [18,19].

### 4.2. Characteristics and Medicinal Properties of Crithmum maritimum

#### 4.2.1. Anti-Inflammatory Compounds of *Crithmum maritimum*

*Crithmum maritimum* is known as sea fennel or rochepatok samphire and is a member of the Apiaceae family. It grows on maritime cliffs and, more rarely, in the sand. Normally, the leaves are used as a condiment or are eaten as salad [14]. Already used in folk medicine, sea fennel seems to be a very promising candidate for both the pharmaceutical and food industry in order to produce new functional products due to its content of vitamin C, iodine, carotenoids, and great amounts of phenolics compared to other species [23,86]. Additionally, *C. maritimum* has gained increasing attention because it is strongly believed to possess antioxidant and antimicrobial activities [11,12,14,87]. The essential oil (EO) of sea fennel contains several volatile compounds such as limonene, α-pinene, sabinene, p-cimene, β-terpinene, β-myrcene, thymol, γ-terpinene, carvacrol, p-cymol, β-ionone, dillapiole, anisaldehyde, β-caryophyllene, carvone, and myristicin [86,88]. Bioactive compounds identified in *C. maritimum* are HCAs and CGA. It is postulated that CGA is produced as a self-defense mechanism during environmental stresses, such as boron and nitrogen deficit or against ROS. CGA is known for several qualities, including antimicrobial, anti-inflammatory, and immune properties [11,12,14,87]. In terms of medical use, the effect of CGA on hypertension has been proven by multiple studies [89,90,91,92]. Several mechanisms have been postulated on how CGA decreases blood pressure. Some of the postulations are the stimulation of nitric oxide (NO) production through the endothelial-dependent pathway, reduction in free radicals by blocking NAD(P)H oxidase expression and activity and, importantly, by the inhibition of angiotensin-converting enzyme [70,93]. Chauhan et al. (2012) demonstrated the anti-inflammatory and immune properties of CGA. Their study revealed the suppression of Th1 cell cytokines such as IL-2 and IL-12 which play a major role and an essential role, respectively, in tolerance in the thymus and the regulation of IFNy and TNFa [94]. Simultaneously, their study revealed an elevation of Th2 cell cytokines such as IL-10 and IL-4, of which principal functions include the negative regulation of Th1 cells, cytokines, and anti-inflammatory response. Thus, CGA seems to exhibit effects that may prove useful in treating/battling/improving different autoimmune or inflammatory diseases such as rheumatoid arthritis and diabetes mellitus that exerts hypoglycemic and hypolipidemic effects [21]. In the latter, CGA seems to mitigate the damaging effects induced by hyperglycemic conditions in both pre-and post-treatment of human hepatocytes cells [95].

*Crithmum maritimum* is also rich in EOs which are proposed to be produced as a self-defense mechanism due to stressful events, and their amount reaches about 0.8% in fruits and from 0.15 to 0.3% in leaves [96]. The EOs mainly comprise monoterpene hydrocarbons and oxygenated monoterpenes. The major oil components are *p*-cymene, *β*-phellandrene, *β*-terpinene, thymol methyl ether, and dillapiole [23]. The monoterpenes, especially the thymol, are believed to play an important role in the odors and taste of the plant [16]. Additionally, a study by Jallali et al. [13] showed that the EOs have antioxidant effects. However, it was low compared to the acetone extract [23]. The oils revealed beneficial protective abilities, indicating that they can protect a lipid matrix from an oxidative event by the formation of hydroperoxydienes (primary oxidation) and by reducing the degradation of these (secondary oxidation) [23,24,36]. Altogether, *C. maritimum* contains several bioactive compounds exhibiting anti-inflammatory and antioxidant properties. Thus, the non-volatile EOs extract rich in hydroxycinnamic acids and flavonoid glycosides, obtained after the hydrodistillation process, have important biological activities, thus endorsing the industrial exploitation of this plant [97].

#### 4.2.2. Antimicrobial Compounds of *Crithmum maritimum*

As mentioned, EOs have antioxidant effects. However, the important role of EO is antimicrobial activity, despite it being less potent than that of synthetic antibiotics. Nevertheless, due to numerous divergent mechanisms of action, they may have the potential to withstand resistant strains of microorganisms [23,24]. Meot-Duros et al. [8,9] have demonstrated that the *C. maritimum* has excellent antimicrobial activity against bacteria such as *Pseudomonas aeruginosa*, *Candida albicans,* and *Escherichia coli* [11,12]. From the *C. maritimum* leaf, they purified and identified falcarindiol, a polyacetylene with several biological activities such as antibacterial, anti-inflammatory, and scavenging activity [11,12,14]. Additionally, they revealed that falcarindiol has antimycobacterial properties against *Mycobacterium tuberculosis* [11,12]. Several studies have also demonstrated cytotoxic properties of falcarindiol against cell lines such as lymphocytic leukemia and human myeloma [12,98,99,100]. These findings have potential for future research in the field of overcoming antibiotic resistance, where *C. maritimum* is an obvious and possible candidate.

### 4.3. Characteristics and Medicinal Properties of Salicornia Species

#### 4.3.1. Anti-Inflammatory Compounds of *Salicornia europaea* and other *Salicornia species*

*Salicornia europaea* is one of the most salt-tolerant species worldwide and belongs to the Amaranthaceae family. Halophytes belonging to the Amaranthaceae family occur in salt marshes and are exposed to excessive environmental salt concentrations as well as physiological drought. Although not restricted to wetlands, these species dominate saline wetlands, such as inland and coastal salt marshes. They grow in coastal regions across mainly the Mediterranean and East Asia, as well as northern European countries including Denmark and Germany. The species is also known as glasswort or marsh samphire in English. The leaves of the plant are used as a salt substitute, to season vegetable, and as a nutritious fermented food, mostly in Korea but also in some European countries. The nutritional profile, the antioxidant capacity, and microbial quality of the produced plants have been evaluated, including minerals and vitamins [101]. This is due to its nutritional and therapeutic importance for treating constipation, obesity, and diabetes [25,54]. It grows in extreme saltwater areas, such as seashores, marsh lands, and salted deserts; therefore, it produces a rich variety of secondary metabolite compounds such as flavonoids, saponins, and alkaloids. These compounds are believed to play a major role in the biological properties of the plant, such as antioxidative, antitumor, antidiabetic, and neuroprotective potential [25]. Botanical extracts from *S. europaea* have also been reported to include saponin compounds, oleanolic acid glucoside, and chikusetsusaponin methyl ester, which have been shown to work in diabetes prevention and as anti-obesity agents [2]. The number of bioactive compounds are relatively higher in matured plants, in comparison to young plants, and the amounts of phenylpropanoic acids and flavonols in *S. herbacea* ethanol extract have shown to increase by 32.6% and 42.4%, respectively, as the shrubs mature [102]. *Salicornia europaea* have been reported to include β-cyanines and isoflavones, which are known for their strong anti-inflammatory and free radical scavenging properties [2]. Flavonoids can be grouped into bioflavonoids, isoflavonoids, and neoflavonoids, and are derived from the same structures, such as flavone, flavonol, and 4-phenylcoumarin, and have been previously investigated for their anti-neuroinflammatory effects [25,103]. In their study, Kim and colleagues [22] show that the isoflavonoid irilin B extracted from *S. europaea* revealed anti-ROS and anti-inflammatory activities in BV-2 microglial cells (in vitro). Thus, their study suggests that irilin B successfully improves the damaging effect of microglia-mediated neuroinflammation and stimulates antioxidative effects. Neuronal death and oxidative stress are some of the hallmarks of neurodegenerative diseases such Parkinson’s. Therefore, the irilin B extract from *S. europaea* may possess anti-Parkinson’s disease-like (anti-PD) potential [25]. Besides *S. europaea*, *S. herbacea* and other *Salicornia* species have also shown to be rich in numerous bioactive compounds, and ethanol extract from the aerial parts of *S. ramosissima* have been reported to include antioxidant alkyl ferulates and coumarin, LDL cholesterol-lowering stigmastanol (syn. sitostanol), and ethyl(E)-2-hydroxycinnamate, which is mostly known for its anti-cancer potential [104]. Sterols and HCAs, such as ferulic acid, have also been detected from the n-hexane extract of *S. ramosissima* [105]. Another succulent halophyte species in the Salicornioideae subfamily is *Arthrocnemum macrostachyum*. Despite their similarities, their total phenolic content seems different, suggesting variability in their chemical composition. AM appears to have a sixfold more active extract than the other species, and thus can exert the highest scavenging activity of reactive species [106]. Flavonoid glycosides including quercetin-3-glucoside and isorhamnetin-3-glucoside can be structurally transformed into minor aglycone molecules, which play a significant role in exerting physiological responses in vivo. Ahn et al. demonstrated that such microbials catalyzed the transformation into quercetin, and isorhamnetin promoted improved anti-inflammatory activity vs. the original source molecules against lipopolysaccharide-induced macrophages [107]. This verifies the anti-inflammatory and antioxidant effects of the *Salicornia* species. Due to these properties, the plants exhibit therapeutic and preventive/protective effects on skin conditions by reducing inflammation of the skin, with the possibility to treat wounds effectively. In some cases, this could be applied in a patch and/or in/on a bandage or simply applied by the hand [106]. In vitro cell assays demonstrated that *Arthrocnemum macrostachyum* possesses a significant concentration-dependent inhibitory performance on matrix metalloproteinase-1 (MMP-1) release by aged fibroblasts. MMP-1 is a collagenase involved in the breakdown of the extracellular matrix [108]. The inhibition of this process reduces, delays, and prevents the breakdown of collagen in the skin and maintains natural collagen levels. Thus, the inhibition of MMP-1 by the halophytic *Arthrocnemum macrostachyum* suggests that the plant can protect the extracellular matrix against damaging outcomes and supply anti-aging effects when applied to the skin. This means that the plant is valuable as a cosmetic for the use of anti-wrinkle cream/extract/agent by reducing or delaying the aging of the skin. The effect could be achieved by applying the invention by several administration routes such as by injection, spray, sponge, and/or directly by the hand [106]. Apart from that, it has been demonstrated that *S. europaea* consists of several other compounds, such as the flavone acacetin [3]. Acacetin, which is an O-methylated flavone, has anti-inflammatory and antioxidant effects that may have a positive effect on sepsis, for example. The explanation for this could be that acacetin strongly inhibits the expression of several proinflammatory cytokines, such as inducible nitric oxide synthase, cyclooxygenase-2, superoxide dismutases, and heme oxygenase-1 [26,60]. Nevertheless, *S. herbacea* also consists of hesperetin, a flavanone found in citrus fruits which showed to have inhibitory effects on microglia-mediated neuroinflammation [3,27]. It was proposed that hesperetin was capable of suppressing MAPK pathways and inflammatory cytokines such as interleukin-1b and IL-6 which are released by activated microglial cells in neurodegenerative diseases, for example [27].

#### 4.3.2. Antimicrobial Compounds of *S. europaea*

As specified earlier, *S. europaea* has anti-inflammatory effects. Nonetheless, its antimicrobial effects have been demonstrated by Essaidi et al. [2], who showed the antimicrobial effect of *S. europaea* against several pathogenic mechanisms, such as *Staphylococcus aureus*, *Escherichia coli*, and *Klebsiella pneumonia*, among others [3]. The extract from *S. europaea* mostly affected Gram-positive bacteria, because they showed the most sensitivity by having a greater inhibition zone than the Gram-negative bacteria [3]. The antimicrobial activity of *S. europaea* is suggested to be attributed to multiple compounds in the plant. These compounds may have bacteriostatic effects because some bacteria are especially sensitive to the extracts from halophytes. Compounds such as phenols are thought to have an impact on this antimicrobial activity. However, this activity is also thought to be associated with components other than phenols; for instance, fatty acids. In addition, Essaidi et al. [2] stated that these extracted compounds from *S. europaea* are potential inhibitors of cytochrome P450 enzymes, such as CYP2D6, CYP1A2, and CYP3A4. This is because *S. europaea* contains flavonols such as quercetin, which have been shown to have inhibitory activity on several cytochrome P450 enzymes. Therefore, it is thought that the inhibition of multiple cytochrome P450 enzymes is related to the presence of phenolic compounds such as phenolic acids and flavonoids in *S. europaea* [3]. The application of mass spectrometry techniques in preclinical investigations and to evaluate the potential biologically active compounds in halophytic plants is encouraged for future studies [109].

## 5. Conclusions

In summary, we have reviewed the current literature and current state of the art for biomedical applications of halophytes such as *A. tripolium*, *C. maritimum*, and *S. europaea*. Conclusively, a number of halophyte species can be used in many applications such as functional food, functional feed, cosmetic products, and finally, as bioactive pharmaceutical compounds. Their properties emphasize their potential for use as medicinal agents such as antibiotics or prebiotics. The massive use of antibiotic treatment has resulted in increased antibiotic resistance, which is one of the most critical treatment problems worldwide. Therefore, there is a growing request for new antimicrobial drugs. Due to the situation, natural derivates and biologically active compounds isolated from plants can be beneficial resources for such new drugs. Halophytic plants are an obvious resource because several studies have proved their antimicrobial effectiveness. Due to the multiple positive effects on health aspects such as antibiotic resistance, regulation of the inflammatory response, and pain analgesia, it increases the need to further investigate the mechanisms and pathways in which these plant species and their secondary metabolites are involved.

## Figures and Tables

**Figure 1 molecules-26-03140-f001:**
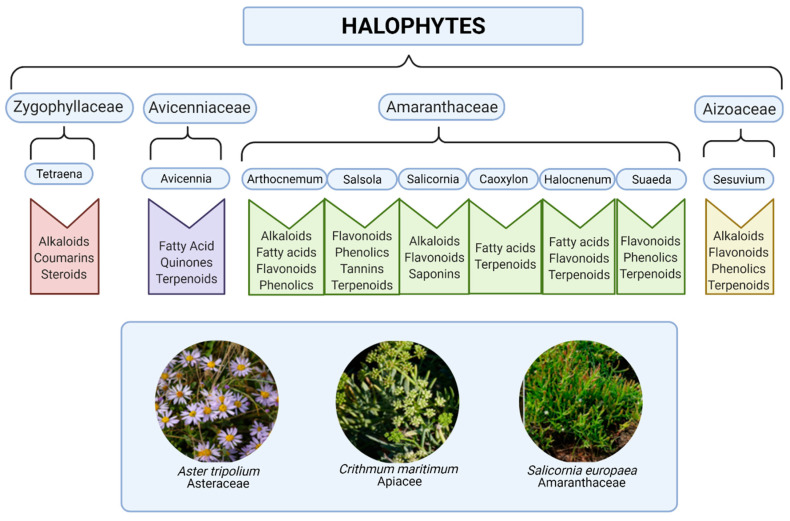
Phytochemical composition of some common coastal halophytes. Created with BioRender.com. (Pictures reproduced with permission [9]).

**Figure 2 molecules-26-03140-f002:**
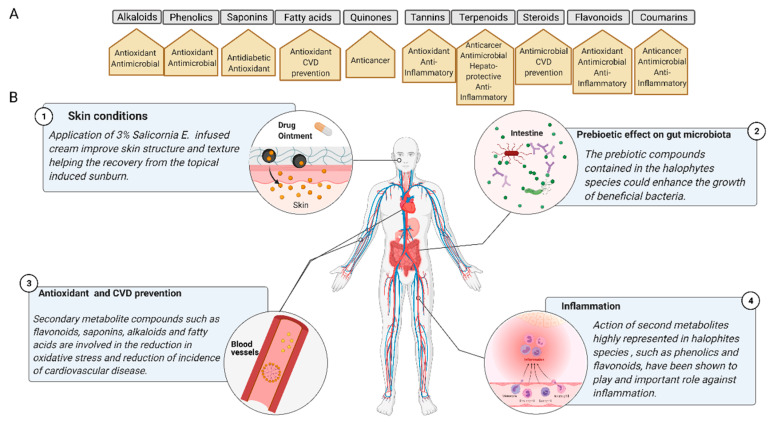
(**A**) Principal physiological processes that involve the action of secondary metabolites present in halophytic species. CVD, cardiovascular disease. (**B**) Main biomedical mechanisms of secondary metabolites’ activities of halophytes. Created with BioRender.com.

**Table 1 molecules-26-03140-t001:** Anti-inflammatory and antibacterial activities of several halophytes based on their bioactive compounds.

	Anti-Inflammatory	Antibacterial	Antioxidative	References
*Aster tripolium*	- Caffeoyl esters (isomer of chlorogenic acid) - Quercetin (flavonoid)	- Caffeic acid (from chlorogenic acid)	- Quercetin (flavonoid)	[11,12,14,16,17,18,19,20,21]
*Crithmum maritimum*	- Chlorogenic acid (hydroxycinnamic acids) - p-Cymene - β-Phellandrene, gamma-terpinene, thymol methyl ether and dillapiole (essential oils)	- Essential oils - Falcarindiol	- Chlorogenic acid (hydroxycinnamic acids)	[11,12,14,21,22,23,24]
*Salicornia europaea*	- Acacetin (flavone) - Chlorogenic acid, rosmarinic acid (esters) - Cinnamic acid, p-coumaric acid, caffeic acid, ferulic acid, sinapic acid, (hydroxycinnamic acids) - Gallic acid, salicylic acid, protocatechuic acid, quinic acid (phenolic acids) - Irilin B (isoflavonoid) - Hesperetin (flavanone) - Galangin isorhamnetin, kaempferol, myricetin, quercetin, rhamnetin, (anthoxanthins flavonol)	- Phenols - Fatty acids	- Tungtungmadic acid - Quercetin - Chlorogenic acid - Caffeoylquinic acid	[3,15,25,26,27,28]

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
