# Peer review of "Pharmacological Insights into Halophyte Bioactive Extract Action on Anti-Inflammatory, Pain Relief and Antibiotics-Type Mechanisms"

_molecules, 2021, doi:10.3390/molecules26113140_

Round 1
Reviewer 1 Report
Pharmacological insights into Halophyte bioactive extract ac-tion on anti-inflammatory, pain relief and antibiotics-type mechanisms
“ High salinity during growth and development also increases the number of free radicals in the plants [2,3].’- authors are advised to rephrase since free radicals cannot be …numbered. Also, authors should bare in mind that increased exposure to free radicals induces redox stress and is generally associated with disease. Maybe a reference to hormesis should be included (doi:10.3390/ijerph17114135).
In this review, there is a focus on three halophytic plant species, Aster tripolium (Jacq.) Dobrocz., Crithmum maritimum L., and Salicornia europaea agg. (illustrated in figure 1) because they are in the focus of a larger project and are being investigated”…”rephrase in order to avoid repetitions
“The biological qualities of these three halophytic plants species can overall be divided into anti-inflammatory and antibacterial according to their bioactive compounds…”- Because the analysis regarding the biological effects is performed based on the individual components in the plants, the antioxidant effect should be also included, since some of the biological compounds listed in table 1 have obvious antioxidant effects (quercetin). This type of effect is also mentioned along the present manuscript so it should be included in table 1
Authors state twice the main focus of the current review (In this review, there is a focus on three halophytic plant species, Aster tripolium (Jacq.) Dobrocz., Crithmum maritimum L., and Salicornia europaea agg. (illustrated in figure 1) “…and The present review will focus on reviewing the current state-of-the-art for biomedici-nal applications of bioactive molecules present in the three halophyte species, A. tripolium, C. maritimum, and S. europaea, …”). Authors should avoid this repetition.
The first tyme an abbreviation is used it should be explained in the text (see HCA at page 4).
“This research showed the polyphenols in the group of flavan-3-ol, epicatechin gallate, and epigallocatechin gallate to have antioxidant properties with 7 and 8 hydroxyl groups, ???re-spectively [“- it is not clear what the authors want to state. Please rephrase for clarity
“However, these pathways might not always be sufficient in all cases, and a non-enzymatic nutritional supplement of antioxi-dants is necessary.” – this section regarding the description of redox imbalance in biological systems is poorly referenced. Also, some references should be included to support the use of vegetal antioxidant supplements in clinical/preclinical practice - doi.org/10.1016/j.phrs.2019.104522, DOI: 10.1016/j.biopha.2019.108612, DOI: 10.3109/03602532.2014.1003649
“Linked to neuroprotection the ROS scavenging by HCAs and their derivatives display antioxidant, anti-collagenase, anti-inflammatory, antimicro-bial and anti-tyrosinase activities, as well as ultraviolet (UV) protective effects, suggesting that they can be exploited as anti-aging and anti-inflammatory agents, preservatives and hyperpigmentation-correcting ingredients [55]” – the phrase is too long and should be rephrased for clarity
“Therefore, Quercetin could thus contribute significantly to the antioxidant de-fences present in blood plasma as reported able to inhibit oxidation of LDL in atheroscle-rotic lesions and thereby be a natural anti-atherosclerotic diet component [14,48].”- the antioxidant and cardio-protective effects of quercetin )are supported by rather old references (before 2000). This should be up-dated DOI: 10.3390/ijms13044839, DOI: 10.1016/j.ejmech.2018.06.053
“ A. trip-olium contains CGA like other halophytic species and contains of caffeoyl esters ??? which are an isomer of CGA.” – rephrase for meaning
“ 4.1.2. Antimicrobial compounds of Aster tripolium” – this title is not appropriate since authors state that no evidence that suggests or clarifies whether A. tripolium possesses antimicrobial properties.
“Crithmum. maritimum is known as sea fennel or rock samphire and is a member of the Apiaceae family. They grow on maritime cliffs…” – replace with IT GROWS!
“ In folk medicine, it is used as a diuretic due to its heaviness on vitamin C, iodine, carotenoids, and great amounts of phe-nolics compared to other species [20,77].” – the above mentioned compounds do not suggest diuretic effects. The plant product is used in folk medicine not as a results of a phytochemical analysis but as a results of practical experience
“ CGA is known for several qualities including antimicrobial, anti-inflammatory, and immune properties” there are also experimental studies proving the hepatoprotective effects of CGA - DOI: 10.25083/rbl/24.2/301.307
General remark – the current reference list includes only 4 titles from 2020-2021 in a total of 95 references. It should be up-dated with a higher percentage of new references
Author Response
Dear Editor and reviewer,
We thank for the rapid review and have below made our best efforts to revise a manuscript based on the reviewers comments. We find that the revised version
“ High salinity during growth and development also increases the number of free radicals in the plants [2,3].’- authors are advised to rephrase since free radicals cannot be …numbered. Also, authors should bare in mind that increased exposure to free radicals induces redox stress and is generally associated with disease. Maybe a reference to hormesis should be included (doi:10.3390/ijerph17114135).
- High salinity during growth and development also increases the level of free radicals in the plants [2,3]. Such potential beneficial dietary factors in small doses and complex combinations (e.g., polyphenols, fibers, polyunsaturated fatty acids, etc.) for lifestyle changes can lead to reduced inflammation and improved health, however, metabolic disturbances are key contributors to disease progression [4].
In this review, there is a focus on three halophytic plant species, Aster tripolium (Jacq.) Dobrocz., Crithmum maritimum L., and Salicornia europaea agg. (illustrated in figure 1) because they are in the focus of a larger project and are being investigated”…”rephrase in order to avoid repetitions
- We have revised the sentense "In this review, there is a focus on three halophytic plant species, Aster tripolium (Jacq.) Dobrocz., Crithmum maritimum L., and Salicornia europaea agg. (illustrated in figure 1) due to their potential use in nutraceutical foods, cosmetics, and also bioactive components for medicinal applications as they contain health-promoting compounds such as minerals, fibers, oils, phenolics and vitamins [13]. "
“The biological qualities of these three halophytic plants species can overall be divided into anti-inflammatory and antibacterial according to their bioactive compounds…”- Because the analysis regarding the biological effects is performed based on the individual components in the plants, the antioxidant effect should be also included, since some of the biological compounds listed in table 1 have obvious antioxidant effects (quercetin). This type of effect is also mentioned along the present manuscript so it should be included in table 1
- We have added the antioxidant effect in table1 and references
Authors state twice the main focus of the current review (In this review, there is a focus on three halophytic plant species, Aster tripolium (Jacq.) Dobrocz., Crithmum maritimum L., and Salicornia europaea agg. (illustrated in figure 1) “…and The present review will focus on reviewing the current state-of-the-art for biomedici-nal applications of bioactive molecules present in the three halophyte species, A. tripolium, C. maritimum, and S. europaea, …”). Authors should avoid this repetition.
The first tyme an abbreviation is used it should be explained in the text (see HCA at page 4).
- We have combined the two sections to a single paragraf:
"The present review will focus on reviewing the current state-of-the-art for biomedicinal applications of bioactive molecules present in three halophytic plant species, Aster tripolium (Jacq.) Dobrocz., Crithmum maritimum L., and Salicornia europaea agg. (illustrated in figure 1) due to their potential use in nutraceutical foods, cosmetics, and also bioactive components for medicinal applications as they contain health-promoting compounds such as minerals, fibers, oils, phenolics and vitamins [14]. The biological qualities of these three halophytic plants species can overall be divided into anti-inflammatory, antioxidant-rich and antibacterial activities, and future use in medicinal practice. The bioactive compounds as listed in table 1 and summarized in section 2."
“This research showed the polyphenols in the group of flavan-3-ol, epicatechin gallate, and epigallocatechin gallate to have antioxidant properties with 7 and 8 hydroxyl groups, ???re-spectively [“- it is not clear what the authors want to state. Please rephrase for clarity
We have rephrased "Rice-Evans et al. [43], investigated the impact of number of hydroxyl groups on antioxidant activity of each molecule compared to vitamin E and proposed a correlation between the number of hydroxyl groups in HCAs, phenolic acids, flavonoids, their respective derivatives, and their antioxidant capacity. Here, the antioxidative capacity of the catechins epicatechin gallate (ECG) and epigallocatechin gallate (EGCG) is due to the molecular presence of 7 and 8 hydroxyl groups, respectively [43]. The latter compounds are well known antioxidants relevant to cancer and neurodegenerative diseases [44,45]. "
“However, these pathways might not always be sufficient in all cases, and a non-enzymatic nutritional supplement of antioxi-dants is necessary.” – this section regarding the description of redox imbalance in biological systems is poorly referenced. Also, some references should be included to support the use of vegetal antioxidant supplements in clinical/preclinical practice - doi.org/10.1016/j.phrs.2019.104522, DOI: 10.1016/j.biopha.2019.108612, DOI: 10.3109/03602532.2014.1003649
We have extended the paragraf and added new very recent references: "However, these pathways might not always be sufficient in all cases, and a non-enzymatic nutritional supplement of antioxidants is necessary. Diet management using plant-based formulations may improve the metabolic status of patients including diabetes mellitus where an increased oxidative stress and chronic low-grade inflammation is observed as consequence of the complex syndrome including long-term alterations of protein and lipid metabolisms [49]. Here researchers found that a plant-based antidiabetic formulation including antioxidants was able to enhance total serum antioxidant defense and improve overall serum redox status and HDL redox function. More than 5000 flavonoids have been identified and are distributed in a wide range of plants and flavonoids possess documented anticancer activity both in animal and cellular model systems [50]. Here, luteolin is an important natural antioxidant which have potent anticancer effects under both in vitro and in vivo conditions."
“Linked to neuroprotection the ROS scavenging by HCAs and their derivatives display antioxidant, anti-collagenase, anti-inflammatory, antimicro-bial and anti-tyrosinase activities, as well as ultraviolet (UV) protective effects, suggesting that they can be exploited as anti-aging and anti-inflammatory agents, preservatives and hyperpigmentation-correcting ingredients [55]” – the phrase is too long and should be rephrased for clarity
We have revised the sentence "Linked to neuroprotection the ROS scavenging then HCAs and their derivatives display antioxidant, anti-collagenase, anti-inflammatory, antimicrobial and anti-tyrosinase activities, as well as ultraviolet (UV) protective effects. This suggests that HCAs can be exploited as anti-aging and anti-inflammatory agents, preservatives and hyperpigmentation-correcting ingredients [59]. "
“Therefore, Quercetin could thus contribute significantly to the antioxidant de-fences present in blood plasma as reported able to inhibit oxidation of LDL in atheroscle-rotic lesions and thereby be a natural anti-atherosclerotic diet component [14,48].”- the antioxidant and cardio-protective effects of quercetin )are supported by rather old references (before 2000). This should be up-dated DOI: 10.3390/ijms13044839, DOI: 10.1016/j.ejmech.2018.06.053
We have rewritten and updated the paragraf: "Quercetin is a very strong antioxidant also found in foods such as apples, onions, and tea [15,19]. Accordingly, quercetin can chelate metals, scavenge oxygen-free radicals, and prevent oxidation of low-density lipoprotein (LDL) in vitro [16]. Therefore, Quercetin could thus contribute significantly to the antioxidant defenses present in blood plasma as reported able to inhibit oxidation of LDL in atherosclerotic lesions and thereby be a natural anti-atherosclerotic diet component or be used in T2DM to achieve adequate glycaemic control [78–80]."
“ A. trip-olium contains CGA like other halophytic species and contains of caffeoyl esters ??? which are an isomer of CGA.” – rephrase for meaning
The paragraf have been refrased: "A. tripolium also contains CGAs like other halophytic species. CGA is a family of esters constructed between certain trans-cinnamic acids and trans-quinic acid [21]. Interestingly though, if CGA isomers are hydrolyzed to quinic and caffeic acids, the latter has shown antimicrobial effectiveness against certain Gram-bacteria."
“ 4.1.2. Antimicrobial compounds of Aster tripolium” – this title is not appropriate since authors state that no evidence that suggests or clarifies whether A. tripolium possesses antimicrobial properties.
We would like to keep the subheading to keep the overall structure in sections. The reviewer is fully correct as we clearly state that our litterature search so far do not support antimicrobial use of this species yet.
“Crithmum. maritimum is known as sea fennel or rock samphire and is a member of the Apiaceae family. They grow on maritime cliffs…” – replace with IT GROWS!
Corrected.
“ In folk medicine, it is used as a diuretic due to its heaviness on vitamin C, iodine, carotenoids, and great amounts of phe-nolics compared to other species [20,77].” – the above mentioned compounds do not suggest diuretic effects. The plant product is used in folk medicine not as a results of a phytochemical analysis but as a results of practical experience
We have rewritten the sentence based on the two references "Already used in folk medicine, sea fennel seems to be a very promising candidate for both pharmaceutical and food industry in order to produce new functional products due to its content of vitamin C, iodine, carotenoids, and great amounts of phenolics compared to other species [22,84]"
“ CGA is known for several qualities including antimicrobial, anti-inflammatory, and immune properties” there are also experimental studies proving the hepatoprotective effects of CGA - DOI: 10.25083/rbl/24.2/301.307
We have rewritten and added the reference "Thus, CGA seem to exhibit effects that may prove useful in treating/battling/improve different autoimmune or inflammatory diseases such as rheumatoid arthritis and diabetes mellitus that exerts hypoglycemic and hypolipidemic effects [20]. In the latter, CGA seems to mitigate the damaging effects induced by hyperglycemic conditions in both pre-and post-treatment of human hepatocytes cells [92]. "
General remark – the current reference list includes only 4 titles from 2020-2021 in a total of 95 references. It should be up-dated with a higher percentage of new references
We have updated our cross-reference search and included multiple recent research and highly relevant review papers. In the revision for the reviewers reccomandations and table1 we have included multiple recent advancements from 2020 and 2021 extending the reference list to 109 references in total. However, the number of relevant references published in 2020-2021 remain relatively low. The major changes are noted in the revised manuscript in red.

Reviewer 2 Report
The manuscript Pharmacological insights into Halophyte bioactive extract action on anti-inflammatory, pain relief and antibiotics-type mechanisms showed a solid review about Salt-tolerant plant species. In my opinion, the manuscript must be published without modification.
Author Response
Dear Editor and reviewer,
The authors thank for the efforts to provide a review in a short time and the nice comments.
On behalf of the authors
Allan Stensballe
Reviewer 3 Report
The authors present a review describing the pharmacological properties of the halophyte extract. The review is well written and gives a good scientific contribution. I believe it can be published in Molecules in its current form.
Author Response
Dear Editor and reviewer,
The authors thank for the efforts to provide a review and the nice comments.
On behalf of the authors
Allan Stensballe
Round 2
Reviewer 1 Report
Authors responded to queries.